# Vector-Transmitted Flaviviruses: An Antiviral Molecules Overview

**DOI:** 10.3390/microorganisms11102427

**Published:** 2023-09-28

**Authors:** Erica Diani, Anna Lagni, Virginia Lotti, Emil Tonon, Riccardo Cecchetto, Davide Gibellini

**Affiliations:** 1Department of Diagnostic and Public Health, Microbiology Section, University of Verona, 37134 Verona, Italy; anna.lagni@univr.it (A.L.); virginia.lotti@univr.it (V.L.); riccardo.cecchetto@univr.it (R.C.); 2Unit of Microbiology, Azienda Ospedaliera Universitaria Integrata Verona, 37134 Verona, Italy; emil.tonon@studenti.univr.it

**Keywords:** antiviral, drug, flavivirus, public health, zoonotic, DENV, TBEV, WNV, YFV, ZIKV

## Abstract

Flaviviruses cause numerous pathologies in humans across a broad clinical spectrum with potentially severe clinical manifestations, including hemorrhagic and neurological disorders. Among human flaviviruses, some viral proteins show high conservation and are good candidates as targets for drug design. From an epidemiological point of view, flaviviruses cause more than 400 million cases of infection worldwide each year. In particular, the Yellow Fever, dengue, West Nile, and Zika viruses have high morbidity and mortality—about an estimated 20,000 deaths per year. As they depend on human vectors, they have expanded their geographical range in recent years due to altered climatic and social conditions. Despite these epidemiological and clinical premises, there are limited antiviral treatments for these infections. In this review, we describe the major compounds that are currently under evaluation for the treatment of flavivirus infections and the challenges faced during clinical trials, outlining their mechanisms of action in order to present an overview of ongoing studies. According to our review, the absence of approved antivirals for flaviviruses led to *in vitro* and *in vivo* experiments aimed at identifying compounds that can interfere with one or more viral cycle steps. Still, the currently unavailability of approved antivirals poses a significant public health issue.

## 1. Introduction

Flaviviruses are positive single-stranded RNA viruses belonging *to the Flaviviridae family* [1]. This viral family is classified into four genera, wherein the *Flavivirus* and *Hepacivirus* genera are related to the onset of clinically relevant human diseases. The Flavivirus genus includes several viruses, including, for example, dengue virus (DENV), Zika (ZIKV), and West Nile (WNV), which are well-known causative agents of human diseases.

These viruses are generally transmitted through the bite of infected arthropod vectors, and over the past seven decades, they have spread widely over the world.

Dengue viruses cause over 3.7 million cases and approximately 2000 dengue-related deaths in 70 countries globally [2].

It is noteworthy that West Nile (WNV) and Zika (ZIKV) viruses were detected in European patients, inducing important clinical impact [3,4] with 1340 locally acquired human cases of West Nile virus, including 104 deaths in Europe [5]. And in 2023, up to 8758 ZIKV cases were reported in the Region of the Americas [6]. ZIKV infection have been linked to severe diseases in adults, including multi-organ failure, meningitis, and encephalitis, and it has been related to death in children with sickle-cell disease and cases of Guillain—Barré syndrome, a progressive polyneuropathy linked to ZIKV infection that occurs in 1/6500 to 1/17,000 people in endemic areas [7]. Other flaviviruses, such as the Usutu virus (USUV), the tick-borne encephalitis virus (TBEV), and the Japanese encephalitis virus (JEV), continue to pose as health risks and are starting to spread around the world [8]. Many factors contribute to flavivirus epidemic potential, including the unique characteristics of their insect vectors, the consequences of excessive and poorly planned urbanization that create ideal breeding habitats for arthropods, and cause vector geographical expansion, climate change, and extensive global travel, facilitating the geographical spread of viruses and vectors [9,10]. These new global conditions determine the expansion of arthropod ranges and their spread into new areas. The flaviviruses can be organized according to transmission route, such as mosquito-borne viruses, tick-borne viruses, and viruses that are unclassified or with unknown vectors. *Aedes* and *Culex* mosquitoes represent the classical vector of mosquito-borne flaviviruses, whereas many different tick species are involved in the transmission of tick-borne viruses [11]. These flaviviruses account for up to 400 million cases per year globally. The major viruses responsible for human infections are DENV, followed by WNV, ZIKV, and YFV, while JEV, TBEV, and Usutu viruses represent an emerging public health risk in specific geographical regions [8].

The epidemiological and clinical characteristics of flavivirus infections affect vaccines and therapeutic drug development. Although many antiviral drugs has been discovered for hepaciviruses, antivirals for the treatment of flaviviruses are not yet available. This discrepancy is related to the large impact of Hepatitis C virus (HCV) infections worldwide; the availability of vaccines for some of the most significant flavivirus, such as YFV and JEV; the large variability of the clinical impact of flaviviral infection from asymptomatic infections to severe disease (as for ZIKV, JEV, TBEV, WNV, and YFV, as mentioned before); and the specific geographic localization of certain flaviviruses restricted to developing countries in Africa and Southeast Asia that lack domestic financial resources and a supply chain [12]. Hence, this review summarizes the current studies on putative drugs and their viral or non-viral targets while also addressing ongoing clinical trials to describe the state of the art characteristics of these items. 

## 2. Biology of Flaviviruses

The knowledge of flaviviridae structure and biology is pivotal to understand the possible targets of anti-Flaviviridae compounds. In fact, in the next paragraphs, we will display the different drugs with anti-Flavivirus activity through the classification of targets and activity mechanisms.

The *Flavivirus* genus shows morphological uniformity characterized by an enveloped virus with an icosahedral capsid. The genome is represented by a single-stranded positive-sense RNA molecule of approximately 11 kb that is translated into a polyprotein in the cell cytoplasm. This polyprotein is cleaved by proteases, releasing three structural proteins: the capsid (C); membrane (M), which is expressed as the precursor (prM); envelope (E) proteins; as well as seven non-structural proteins, namely NS1, NS2A, NS2B, NS3, NS4A, NS4B, and NS5 [13] (Figure 1).

In terms of homology, some genome regions are more conserved between flavivirus genomes, such as the *NS3* and *NS1* genes [15,16,17,18], while others, such as C protein, show the lowest homology, making these proteins less promising as targets for broad-active antiviral drugs, although structural, biochemical, and functional properties with other flaviviruses are fully conserved [18].

Cell infection is mediated by interactions between viral E proteins and cell receptors specific to one or more Flaviviridae [19,20]. For a comprehensive list of viral host receptor, refer to Table 1.

These viruses are released into the cell cytoplasm by endocytosis, and their genome is translated by ribosomes into a viral polyprotein, which is cleaved via host and viral proteases into structural and non-structural viral proteins. After the release of these cleaved proteins, viral genome replication occurs at the endoplasmic reticulum. The viral replication complex is composed of several viral proteins, each one with its own function, such as viral RNA-dependent RNA polymerase (RdRp) NS5, which governs viral replication. Virion assessment and maturation take place in the endoplasmic reticulum, followed by processing in the Golgi apparatus, and then the exocytosis of mature virions from infected cells. Each of these stages of infection is discussed in more detail in the following paragraphs.

## 3. Transmission and Pathology

Flavivirus transmission generally occurs through vector insect bites, usually mosquitoes or ticks. The incubation period ranges from 3 days to 2–4 weeks, as in the case of Murray Valley encephalitis virus (MVEV). Interestingly, several cases of flavivirus infections are often clinically asymptomatic, but it is possible to note fever, headache, skin rash, and nausea, which generally occur without clinical consequences in symptomatic cases, except when they represent the first stage of severe hemorrhagic fever or neurological damage, which can lead to death, depending on the flavivirus [70,71,72,73].

The possibility of a serious symptomatic evolution depends on virus type, specific immunological situation, the presence of co-morbidities, previous heterologous infections, and the age of the patient. Although transmission mainly depends on insects, some flaviviruses can also be transmitted from human to human. While it is not possible for most flaviviruses to demonstrate placental infection and fetal involvement, ZIKV, for example, has been proven to cross the placental barrier and elicit teratogenic effects in developing fetal tissues, especially in neurological tissues. In addition, ZIKV is detectable and persists in human semen and cervical mucus, reinforcing the possibility of vertical transmission and indicating the possibility of sexual transmission. A consistent viral load for not only ZIKV, but also DENV, YFV, and WNV is also detectable in breast milk as well as in saliva (ZIKV), whereas the blood and urine viral load is detectable at variable concentrations depending on the flavivirus, thus suggesting alternative transmission routes.

## 4. Viral Targets and Drugs with Antiviral Activity

Treatments for viral infections must inhibit the viral cycle in order to prevent the formation of new viral progeny and halt the increase in viral load. In addition, the target must be stable and not prone to mutations that could reduce or suppress antiviral activity. These two features are essential to tackle the increase in viral load and the onset of resistance that are related to the failure of treatment. The importance of a stable genome is exemplified by HCV since viral RNA-dependent RNA polymerase (RdRp) does not carry out proofreading in hepaciviruses or flaviviruses, thus producing quasi-species during infection, allowing for efficient immune evasion and the production of new defective, non-functional virions [74,75]. Viral targets are related to molecules that inhibit or interfere with viral proteins with enzymatic activity or that can alter proteins through a structural non-enzymatic action in order to render them ineffective, for example, the maturation and formation of new virions or virus/cell-binding. This procedure was used during drug development for the Human immunodeficiency virus (HIV) and HCV; reverse transcriptase and protease inhibitors were approved for the treatment of HIV, thereby blocking the enzyme’s activity [76]. HCV antivirals, such as NS3-NS4A protease inhibitors, NS5A inhibitors, and NS5B polymerase inhibitors, were successfully developed [77]. Since HCV is related to flavivirus (both belonging to the same family *Flaviviridae*), these target and antiviral mechanisms could be exploited for the study of antivirals against flaviviruses. In this review, we analyze viral targets in different stages of the replication cycle (Figure 2).

### 4.1. Entry Inhibitors

The first step of the viral replication cycle is the entry of the virus into the cell through the cell–virus interactions at the cell membrane level. For flaviviruses, their entry strategy is based on maximizing virion concentration on the cell surface. To achieve this, flaviviruses recognize cell membrane targets, including glycosaminoglycans. This interaction, mediated by domain III of E proteins, is non-specific and characterized by low affinity, but it guarantees the attachment of a high concentration of virions to the surface [80]. Van der Schaar and collaborators [81] demonstrated that DENV virion particles can spread across the cell membrane and simultaneously bind to specific receptors localized in regions with a high presence of clathrin, called clathrin-rich regions. Notably, the interaction between the viral E glycoprotein and the cell receptors not only induces the first phase of the viral cycle, but also activates of signal transduction pathways, influencing viability and cell proliferation, as well as cytoskeleton structure regulation (Figure 3). Cellular receptors are virus-specific (Table 2) and recognize viral envelope E glycoproteins through domain III [80] or (rarely) through domain II [82].

Flavivirus E protein is a surface membrane protein involved in host cell receptor-binding functions. It mediates the fusion of the viral envelope with endosomal membranes and is necessary for the proper virion assembly, maturation, and secretion [20].

Receptor-specific binding and clathrin-coated pockets determine the invagination of the plasma membrane and consequent endocytotic vesicle formation. In the endosome, the virus undergoes conformational changes, including trimerization of E protein, a basic step for membrane fusion [83]. Nucleocapsid is released in the cytoplasm, where capsid rearrangements take place with the dissociation of capsid proteins from viral RNA [84,85,86].

Studies assessing molecules involved in receptor inhibition or co-receptor-binding, as well as entry and fusion to inhibit the first phase of the viral cycle do, not have a primary application in virology. Some compounds, such as T-20/enfuvirtide and maraviroc, are used in HIV treatment, but these molecules do not have the antiviral impact of drugs that inhibit retro-transcriptase, or protease or integrase activity. More specifically, T-20/enfuvirtide is a gp41 membrane-proximal external region (MPER)-derived peptide that inhibits the fusion between virus and cell membrane. This feature has suggested [87] that an appropriate modification of flavivirus-derived stem peptides might result in a good inhibitor of the flavivirus fusion process [87].

Different classes of viral E glycoprotein-targeting drugs were identified to interfere with the first step of infection using *in vivo* cell culture and animal models (mice and non-human primates) (Table 2).

**Table 2 microorganisms-11-02427-t002:** Flavivirus entry inhibitors.

Target	Drug	Viral Specificity	Study Stage	Ref.
E	Z2	DENV, YFV	*In vitro*	[88,89]
		ZIKV	*In vivo*	[88]
	DN59	DENV-2	*In vitro*	[90,91]
		WNV	*In vitro*	
	P5	JEV, ZIKV	*In vivo*	[92]
	DET2 and DET4	DENV-2	*In vitro*	[93,94]
	Dipeptide EF	DENV	*In vitro*	[95]
	JBJ-01-162-04	DENV, JEV, WNV, ZIKV	*In vitro*	[96]
	mAb513	DENV	*In vivo*	[97]
	2D22	DENV	*In vivo*	[97,98,99]
	ZIKV-Ig	ZIKV	Phase 1 clinical trial	[100]
	TY014	YFV	Phase 1 clinical trial	[101]
	Tyzivumab	ZIKV	Phase 1 clinical trial	[101]
	MGAWN1	WNV	Clinical trial withdrawn for low enrollment	[101]
Viral entry	Geraniin	DENV-2	*In vivo*	[102,103]
	Palmatine	WNV, DENV-2, JEV, YFV, ZIKV	*In vitro*	[104]
	Prochloroperazine (PCZ)	DENV, JEV	*In vitro*	[105]
	Daptomycin	JEV	*In vitro*	[106,107,108]
		Puerto Rico ZIKV		
	Nanchangmycin	CHIKV, DENV, WNV		
	Erlotinib, Sunitinib	DENV	*In vivo*	[109]
		WNV, ZIKV	*In vitro*	[109]
	25-hydroxylcholesterol	DENV, YFV, WNV	*In vitro*	[110]
		ZIKV	*In vivo*	
	Chloroquine	ZIKV	*In vivo*	[111,112]
		DENV	Phase 2 clinical trial failed (no viremia reduction)	[107,113]
	Niclosamide	DENV, WNV, YFV, JEV	*In vitro*	[114]
		ZIKV	*In vivo*	

#### 4.1.1. Synthetic Peptides

One approach used to develop flavivirus treatment is synthetic peptides designed to bind to viral E glycoprotein or inhibit processes, such as receptor binding or fusion. The development of peptide drugs has become more successful because of their improved safety compared to other classes of antiviral molecules and antibody-based antiviral drugs. Many synthetic peptides have been tested, and we describe some of the most significant inhibitors in the following paragraphs.

Z2 peptide is a synthetic peptide derived from the conserved stem region of ZIKV E glycoprotein. This peptide binds to viral E protein, causing virion damage, disrupting the integrity of the ZIKV membrane and interrupting fusion, thus releasing the RNA genome [88,89]. Z2 peptide exhibited strong inhibitory activity in vitro against ZIKV, YFV, and DENV. Interestingly, the Z2 treatment of ZIKV-infected A129 and AG6 mice protected approximately 70% of mice from death, significantly reduced the viral load, and eliminated neurological symptoms. Importantly, this compound was also tested in ZIKV-infected pregnant C57BL/6 mice, resulting in a reduced viral load in placentas and fetal CNS, thus demonstrating the ability of Z2 to overcome the placental barrier and protect pregnant mice from neurological damage from vertical viral transmission [88]. The mechanism of flavivirus inhibition by Z2 peptide is related to damage induction in the viral structure with loss of the RNA genome and virion alteration through the disruption of membrane bilayer structures [115]. No side effects were noted with Z2 peptide administration.

Similarly, DN59 peptide is a 33-amino-acid (aa) mimetic peptide corresponding to the membrane-interacting stem region of DENV-2 (aa 412–444) E glycoprotein; its involvement in structural rearrangements during fusion was tested. DN59 is responsible for >95% of viral plaque reduction during DENV and WNV challenge [90,91].

P5 peptide is derived from helix 2 of the JEV E protein stem region. It blocks virus infection through non-specific and hydrophobic membrane binding, followed by interaction with E proteins during fusion [92]. P5 demonstrated antiviral activity against JEV and ZIKV *in vitro*. In JEV-infected mice, P5 treatment resulted in 67% survival and a reduction in viremia and inflammation in mouse brain. P5 can also reduce histopathological damage in the brain and testes in ZIKV-infected AG6 mice [92].

Other synthetic 10-mer peptides, DET2 and DET4, target domain III of DENV-2 E proteins, and their antiviral performances were successfully tested against DENV infection. Transmission electron microscopy (TEM) images revealed that their antiviral effect is due to their ability to induce structural abnormalities and conformational alterations of viral E glycoproteins [93,94]. DENV-2 infection was inhibited in LLC-MK2 cells, especially by DET4 peptide.

Dipeptide EF is a peptide that targets the hydrophobic pocket of E proteins, thereby interfering with the membrane fusion process. Dipeptide EF strongly reduces DENV viral genome replication with all DENV serotypes, although the effects on DENV-2 are more important [95].

#### 4.1.2. Monoclonal Antibodies

Antibodies against viral E glycoprotein are important for inhibiting viral infection. Several viruses were tested with neutralizing antibodies that interfere with virus/cell binding. This approach was employed against other viruses, including SARS-CoV-2, where the monoclonal antibodies (mAbs) are a useful treatment for infected patients [116].

Two mAbs have been developed as potential antivirals. The first mAb, Ab513, specifically binds to the A strand of the E glycoprotein domain III of DENV-4, but it can bind with high affinity to neutralize several DENV genotypes [97]. Prophylactic administration in mouse models subsequently infected with DENV-2 significantly decreased the viral load and increased survival, confirming its protective activity against DENV in the presence of heterologous enhancing antibodies.

The second monoclonal antibody, 2D22, binds to E protein in dimer conformation and abolishes E conformational changes, which are essential for fusion [98]. In the mouse model AG129, 2D22 mAb is highly effective against DENV, preventing lethal effects of virus and preventing the development of antibody-enhanced vascular leakage [97,98,99].

Human anti-ZIKV immunoglobulin (ZIKV-Ig) has been proposed as a ZIKV treatment; it is currently undergoing a phase 1 double-blind, randomized, placebo-controlled study [100]. Other therapeutic antibodies were developed [117] for the neutralization of YFV, ZIKV, and WNV infection, such as TY014, Tyzivumab, and MGAWN1, respectively. These three antibodies were in trial analysis, but Tyzivumab has discontinued in one of two clinical trials in phase 2 because of difficulties in the cohort assessment due to a lack of ZIKV infected patients, whereas TY014 has successfully completed the phase 1 of clinical trial. Similar to Tyzivumab, even MGAWN1 had been withdrawn from the study due to low enrollment even though phase 1 has demonstrated that this recombinant humanized monoclonal antibody is safe and well tolerated in healthy subjects [101]. There are several drawbacks to monoclonal antibody therapy, including restricted availability and accessibility, high cost, the requirement for early intervention, and the danger of side effects [118]; in particular, with mAbs, the possibility of antibody-dependent enhancement (ADE) exacerbation must be considered, as mAbs against RSV, MERS, and SARS-CoV-2 has been shown to induce ADE at lower concentrations [119].

#### 4.1.3. Synthetic and Natural Compounds

Several groups challenged some classes of synthetic peptides with a putative antiviral activity. Notably, Li and coworkers [96] described a series of cyanohydrazones with inhibitory activity against DENV and subsequently against other flavivirus, including ZIKV, WNV, and JEV. In particular, JBJ-01-162-04 cyanohydrazone showed the best antiviral activity in an *in vitro* cell model targeting a conserved pocket of n-octyl β-D-glucoside of E glycoprotein, which inhibited fusion and viral entry into the cell [96].

Among natural compounds, palmatine and geraniin, demonstrated their anti-flaviviral activity against DENV-2 (for geraniin), ZIKV, and JEV (for palmatine). Palmatine is a plant metabolite, a protoberberine alkaloid derived from *Coptis chinensis*, and was selected by a molecular docking analysis [120], while geraniin, which is extracted from *Nephelium lappaceum,* belongs to a group of hydrolysable tannins and can reduce viral infectivity in Vero cells, and in BALB/c mice, it reduces the liver damage even though a viral load decline is not significant in serum [102,103].

Treatment with palmatine of infected Vero cells exhibits the inhibition of protease activity, non-detectable cytotoxicity, and viral suppression. Actually, palmatine efficacy is demonstrated for WNV, DENV-2, and YFV [104].

#### 4.1.4. Host Function Target

Interestingly, some groups have indicated the antiviral activity of molecules, not through direct effects on viral targets, but rather through host functions.

Prochloroperazine (PCZ) is a well-known dopamine D2 receptor inhibitor that interferes with viral entry. *In vitro* studies have shown that PCZ is a strong inhibitor of DENV and JEV infection. Additionally, DENV is not inhibited in D2-receptor-knockdown shD2R-N18 cells, suggesting a biological role of the D2 receptor in viral infection. Intriguingly, PCZ can also alter clathrin organization in the cells, thus disrupting clathrin-mediated endocytosis. This mechanism is also used by daptomycin, a lipopeptide that disrupts phosphatidylglycerol-rich membranes, and the polyether nanchangmycin, which is produced by *Streptomyces nanchangensis.* Nanchangmycin inhibits DENV, chikungunya (CHIKV) and WNV, while daptomycin acts on JEV and the Puerto Rico ZIKV strain [106,107,108].

Two anticancer drugs, erlotinib and sunitinib, have demonstrated antiviral activity against WNV, DENV, and ZIKV [109]. Sunitinib is a receptor tyrosine kinase inhibitor used for treating gastrointestinal stromal, renal, pancreatic, neuroendocrine, and meningioma tumors. Their mechanism of action is multitargeted: sunitinib inhibits the phosphorylation of many receptors, including PDGFRs, VEGFR receptors, and c-kit. Erlotinib is an epidermal growth factor tyrosine kinase inhibitor used to treat lung and pancreatic tumors. Treatment with these two molecules inhibited DENV infection in murine models but did not prevent neurological damage because of the low permeability of the blood–brain barrier (BBB) to erlotinib and sunitinib. Their antiviral mechanisms are likely related to the inhibition of viral entry and the formation of virions via large kinase derangement. One of the major challenges in developing flavivirus antivirals is the availability of drugs that can cross the BBB to treat neurological diseases. Due to the BBB’s limited permeability, several antiviral compounds with promising *in vitro* characteristics were subsequently found ineffective *in vivo* [121].

Among the drugs that inhibit viral fusion, 25-hydroxylcholesterol (25HC) can inhibit DENV, YFV, WNV, and ZIKV by affecting viral internalization. This mechanism is related to the ability of 25HC to modulate lipid metabolism, thus inducing a block between the virus and the cell membrane. This drug was tested in BALB/C and A129 mice infected with ZIKV; the treatment reduced mortality and viral load. ZIKV inhibition was also clearly noted in organoids, but interestingly, 25HC has a consistent ability to inhibit neurological alterations and prevent ZIKV infection in the fetal brain when used in pregnant mice [110]. Endosome acidification is an early step of viral replication that is essential for genome release into the cytoplasm. Chloroquine is a well-known antimalarial molecule that has *in vitro* antiviral activity against several viruses [111,112], but its antiviral effect *in vivo* has not been confirmed. Chloroquine was assayed successfully in mice and primate models of ZIKV and DENV. However, two phase 2 chloroquine trials failed to detect a reduction in viremia in patients with DENV, possibly because chloroquine may not reach inhibitory concentrations inside the reticuloendothelial cells where DENV replication is thought to occur. When compared to a placebo, chloroquine was associated with a higher rate of adverse events; however, these were generally moderate. Chloroquine had no effect on the amplitude of cytokine or T cell responses to DENV infection [107,113].

Niclosamide is an anthelmintic drug that was approved for taeniasis treatment. It inhibits mitochondrial ADP phosphorylation and is effective against DENV, WNV, YFV, ZIKV, and JEV [114]. Niclosamide was tested in animal models and in humanized chick embryo system, in which ZIKV replication was reduced, and the central nervous system was somewhat protected by multitarget action, that is, endosomal deacidification and the derangement of NS2B-NS3 complex formation. This multitarget action, both on host target and on viral target, may turn out to be important in case of viral target mutation, as it maintains efficacy on the endosomal deacidification.

## 5. Viral Replication, Polyprotein Synthesis and Processing

After viral genome release, flaviviral RNA undergoes translation by ribosomes associated with the rough endoplasmic reticulum (RER). Viral RNA produces a unique polyprotein, which is released and attached to the internal RER membrane. Subsequently, the proteolytic cleavage, which is performed by viral NS3 and host proteases, disengages structural and non-structural viral proteins, which can form complexes and drive replication organelle formation. NS4A, NS4B, and NS1 play an essential role in replication organelle formation through ER membrane remodeling [122,123,124] (Table 3)

### 5.1. NS1 Targeting

NS1 is a multitasking non-structural viral protein with a molecular weight of 46-55 kDa that plays different roles in the viral replication cycle, virion assembly, viral pathogenesis, and immune evasion [132,196,197,198]. It exists in many oligomeric forms and is found in various cellular locations. Intracellular NS1 is required for virus replication and has been shown to co-localize with dsRNA and other replication complex components; additionally, it can cause complement-mediated immunological suppression and has the ability to alter membrane lipids [199].

In addition, NS1 protein retains high levels of identity and similarity between flaviviruses [16,200], which make it an excellent candidate as an antiviral drug target. In particular, glycan addition to NS1 occurs through the host oligosaccharyl transferase complex into the ER lumen and is essential for NS1 functionality [201]. Moreover, NS1 is detectable in its secreted hexamer soluble form, and in a DENV viral model, it displayed several pathogenetic mechanisms contributing to viral enhancement, vascular barrier disruption, proinflammatory cytokines increase, and immune evasion.

Several drugs with anti-NS1 activity were identified using biopanning assays, an affinity selection technique, with a phage-displayed peptide library [125,126,202]. In particular, Songprakhon as well as Sun and colleagues [125,126] described four 12-mer peptides (peptides 3, 4, 10, and 11) that bind to NS1. These peptides reduced DENV serotypes to different extents within an *in vitro* Huh-7 cell model at millimolar concentration.

In silico molecular docking screening for deoxycalyxin-A, a flavonoid, was carried out to test its ability to target ZIKV NS1 and to predict its high affinity for ZIKV NS1 binding [127]; however, further *in vitro* and *in vivo* studies are needed to confirm this in silico finding.

mAbs have also been designed to target NS1. mAb AA12 shows significant efficacy against African and Asian lineage strains of ZIKV in Stat2^−/−^ mice [128]. More recently, Biering and colleagues [129] analyzed a promising monoclonal antibody called mAb 2B7, which recognizes NS1 viral protein of ZIKV, WNV, and DENV. This binding, mediated by the wing domain of NS1, is able to block endothelial disfunction caused by NS1 action and protect from downstream disorders.

Research is also underway to investigate the unusual sugar presence on the N-terminal side of NS1. Courageot et al. [130] demonstrated reduced viral production after cell treatment with castanospermine and deoxynojirimycin, both of which are α-glycosidase inhibitors, and similar results were obtained by Wu and coworkers [131], who revealed reduced NS1 with consequent low virion production through N-nonyl-deoxynojirimycin treatment. Additional studies on the efficacy of a pro-castanospermine drug (celgosivir) did not show specific effects on viral reduction but revealed a potential “symptomatic” efficacy in the reduction in the severity of dengue clinical manifestations [132,133].

### 5.2. NS2A Targeting

Flavivirus NS2A is a membrane-associated, small, hydrophobic protein involved in RNA replication. NS2A binds to the 3′ untranslated region (UTR) of viral RNA as well as other replication complex components with excellent specificity. NS2A also has a role in influencing the host-antiviral interferon response and virus particle assembly/secretion [203].

To our knowledge, no drugs have been yet tested that target flaviviral protein NS2A. Due to its lipid interactions and involvement in immune regulation and calmodulin binding ability, investigating treatment with competitive molecules for calmodulin-binding and lipid interaction should be tested.

Only a few studies based on the use of RNA interference have identified some short hairpin RNA (shRNA) and small interfering RNA (siRNA) molecules, which are specific for NS2A and are able to have *in vitro* antiviral effect against JEV, albeit to a much lesser extent than the shRNA and siRNA molecules directed against genes encoding structural proteins and against NS1 [134,135].

### 5.3. NS2B-NS3 Targeting

NS3 has a molecular weight of 69 kDa and is a highly conserved protein with two domains: a protease with a trypsin-like serine domain located on the NS3 N-terminus for polyprotein cleavage and a helicase with an NTPase domain in the C-terminus, which is involved in viral genome RNA replication.

Notably, the protease function of NS3 requires viral NS2B as a cofactor. The complex formed by NS3 and NS2B is essential for polyprotein processing. This important activity in viral replication makes NS2B-NS3 a potential putative antiviral molecule.

Several drugs have been proven as effective molecules through *in vitro* studies. In particular, peptidomimetic inhibitors, niclosamide (as indicated above), novobiocin, and temoporfin are considered NS2B-NS3 inhibitors.

Novobiocin is an antibiotic that inhibits bacterial DNA gyrase, and it can interact with the binding pocket of ZIKV NS2B-NS3. In cell cultures, novobiocin strongly inhibits ZIKV and DENV replication, while in mice models, novobiocin significantly decreased the viral load and overall survival in the treated cohort [136,137].

Temoporfin was selected as an NS2B-NS3 inhibitor using high-throughput screening, a method that has dramatically improved pharmacological research due to its rapid and effective selection of new compounds. Temoporfin can bind to the NS3 domain recognized by NS2B, thereby halting polyprotein processing. *In vitro,* DENV, YFV, WNV, and JEV were inhibited by temoporfin, while the ZIKV viral load was strongly inhibited in BALB/c model experiments. In addition, this drug can significantly prevent neurological manifestations in A129 mice infected with ZIKV [114].

A niclosamide derivative, JMX0207, effectively inhibits NS2B-NS3 interactions and significantly inhibits DENV-2 and ZIKV viral replication. This molecule also shows ZIKV infection reduction in 3D mini brain organoids and in a ZIKV animal model [138].

An FDA-approved drug used against HIV and HCV infection, nelfinavir, has been proposed as an NS2B-NS3 protease inhibitor through a MM/GBSA-based binding free energy analysis. It is a peptidomimetic compound that showed low antiviral activity against DENV-2 and CHKV [139].

Novel carbazole derivatives designed with at least one amidine, as well as compound 4 (a carbazole derivative) demonstrated biochemical and cell-based inhibitory activity *in vitro* against ZIKV, inhibiting NS2B-NS3 protease activity [140]. Compounds 14 (C_30_H_25_NO_5_) and 15 (C_34_H_23_NO_7_S_2_), two non-peptide molecules with significant inhibitory effects on the DENV NS2B-NS3 protease, showed moderate *in vitro* antiviral activity against DENV [141]. A virtual screening pipeline showed that NSC135618 significantly inhibited the DENV-2 protease function and inhibited the viral replication of DENV, ZIKV, WNV, and YFV *in vitro* [142].

Through the high-throughput screening of a chemical compound library applied on a whole-virus proteome, ST-610 was identified to be a potent small molecule acting on the DENV NS3 helicase domain. Experiments performed *in vitro* and *in vivo* confirm the ability of ST-610 to avoid NS3 binding with viral RNA, although NS3 still retains its nucleoside triphosphate activity. ST-610, thanks to its non-toxicity and great efficacy in reducing viremia, is an excellent candidate as a treatment for DENV infections [156].

Aprotinin, a well-known drug used during cardiopulmonary bypass to reduce bleeding, is an inhibitor ligand of the NS3pro and NS2B complex of WNV and DENV. It acts through conformational changes and specifically binds to the NS3pro pocket using its antiparallel β-sheet and has *in vitro* antiviral activity against DENV [144,145]. Other inhibitor ligands of the NS3pro/NS2B complex of WNV are substrate analogs, which take advantage of cation-p interactions between P1-Arg residue and the inhibitor benzoyl cap on the NS3pro structure, thus changing its conformational state and ligating the complex, thereby stabilizing NS2B with high efficiency [204,205].

ZP10 (theaflavin-3,3′-digallate), a natural compound derived from black tea, was predicted to bind to critical residues at the proteolytic cavity of NS2B-NS3 proteases, thus inhibiting polyprotein processing. This compound inhibits *in vitro* ZIKV replication in a dose-dependent manner [143].

An allosteric small-molecule inhibitor, NSC157058, interfered with NS2B folding in modeling experiments on WNV and ZIKV. Its antiviral effect was investigated on SIL mice, in which ZIKV viremia was reduced [158].

Hydroxychloroquine (HCQ), a derivative of chloroquine and a U.S. Food and Drug Administration (FDA)-approved drug used to treat patients with autoimmune diseases and malaria, also with pregnant people, was tested in ZIKV-infected placental trophoblast cells and pregnant mice, resulting in a reduced viral load and placental damage. Its antiviral activity is multitargeted: HCQ binds to the NS2B-NS3 binding site impeding autophagosome–lysosome fusion and inhibiting autophagy [146,147].

Methylene blue and erythrosin B were found to be orthosteric inhibitors that significantly inhibit ZIKV and DENV-2 NS2B-NS3 protease activity, the replication of multiple ZIKV strains, and DENV-2 *in vitro* antiviral assays [138,149,150]. Methylene blue inhibited viral replication in primary neural and placental cells and in 3D mini-brain organoids that are relevant to ZIKV pathogenesis. Animal model studies confirmed that methylene blue treatment significantly improved the survival rate of ZIKV-infected mice [148].

Ivermectin, an anthelmintic drug, exerts *in vitro* antiviral activity against YFV, WNV, and DENV in the early stages of infection by inhibiting the NS3 helicase domain [151]. Additional antiviral effects are caused by interfering with ivermectin to affect importin (IMP)-α/β1-mediated import. However, Ketkar and coworkers [152] did not find the same effects in ZIKV mice models, suggesting the need for further research. A phase 2/3 trial is in progress, but to our knowledge, no results are currently available.

Suramin, a well-known anti-parasitic drug, which has been shown to have antiviral activity, through a molecular beacon helicase assay and subsequent counter screen experiments, was identified as having a good effect in reducing DENV NS3 helicase activity [157], with the data subsequently confirmed by Albulescu and colleagues, which revealed a reduction in viral cytopathic effect in cell culture. They detect a reduction in intracellular ZIKV RNA in a dose-dependent manner, along with a decrease in the progeny viral titer [206].

The quantitative high-throughput screening of NS2B-NS3 ZIKV protease inhibitors showed that bortezomib has strong *in vitro* anti-ZIKV activity [106]. In a self-cleavage screening assay, bortezomib indirectly inhibited *in vitro* ZIKV and DENV replication by increasing NS3 ubiquitination and degradation [154]. In the initial docking analysis of approximately 250,000 compounds that interact with the NS2B-NS3 protease complex binding-site of flaviviruses, Pathak and colleagues [155] applied the pharmacophore anchor model to obtain the best chemical groups for anchor interaction. From this analysis, two HCV antivirals, asunaprevir and simeprevir, showed potent *in vitro* anti-ZIKV activity.

### 5.4. NS4A and NS4B Targeting

NS4A is a 16 kDa transmembrane ER resident protein, and it is involved in cellular membrane modeling, antagonizing host interferon response, and inducing autophagy, and allows for viral replication [207]. Drugs targeting NS4A are currently being evaluated, including compound-B and SBI-0090799, which are active *in vitro* against DENV and ZIKV by preventing NS4A involvement in replication complex formation [159,160].

Unlike NS4A, NS4B is often used as a molecular target due to its role in flavivirus replication and in dampening the host’s immune system [208]. NS4B is a multi-transmembrane protein located in the endoplasmic reticulum membrane, where it plays an important role in the formation of the DENV replication complex by binding with NS3 [209].

Studies have suggested NS4B as a pivotal target for putative antivirals [210]. In particular, JNJ-A07, its analog (JNJ-64281802), and JNJ-1802 antivirals were indicated as useful molecules for DENV treatment through the inhibition of NS3-NS4B heterodimerization in the replication complex.

JNJ-A07 prevents NS4B-NS3 interaction [161], thus inducing a conformational change in the cytosolic loop of NS4B. It can also exert its antiviral effects on a large panel of DENV isolates at nanomolar to picomolar concentrations, thus indicating JNJ-A07 as a pan-serotype DENV antiviral. In addition, JNJ-A07 significantly decreased the viral load in AG129 mice. An analog of JNJ-A07, namely JNJ-64281802, was registered for a phase 2 randomized, double-blind, placebo-controlled clinical trial to investigate its efficacy in DENV prophylaxis in healthy individuals and for DENV therapy in infected patients, but no results have been published. JNJ-1802 is a compound similar to JNJ-A07 that shares the same target. This molecule exhibited antiviral activity at nanomolar concentrations and inhibits the replication of all DENV serotypes, JEV, WNV, and ZIKV *in vitro*. In mouse and non-human primate models, this drug can inhibit DENV [211]. A phase 1 clinical trial of JNJ-1802 was successfully completed in humans, and it was safe and well tolerated in healthy individuals [212] (NCT05201794) as well as for patients with confirmed DENV (NCT04906980) [162].

NITD-688 was reportedly an NS4B-targeting drug, but the exact mechanism is being investigated. Nuclear magnetic resonance research has demonstrated that it binds to NS4B, and that two specific mutations in this viral protein (T215A and A222V) abolish this binding. The mechanism is potentially different from that of JNJ-A07 due to differences in the resistance mutation profile, which suggests peculiar inhibitory activity. This drug reduced DENV viremia in AG129 mice and was well tolerated in pharmacokinetics studies in rats and dogs [163]. NITD-688 showed antiviral activity against all four DENV serotypes, and it is considered a possible candidate for preclinical studies of DENV treatment.

Manidipine, a calcium channel inhibitor used in hypertension treatment, demonstrated significant antiviral activity against JEV, DENV, ZIKV, and WNV *in vitro*, and protected JEV-infected mice from brain damage although, viremia was not affected. Its mechanism of antiviral activity is not clear, even though mutant analysis revealed that a single mutation in the transmembrane domain of NS4B caused the failure of manidipine’s antiviral-related effects, thus suggesting the involvement of NS4B [164].

Two drug-validated compounds, which are known to have inhibitory activity on Abl and Src kinases, namely AZD0530 and dasatinib, were tested for their abilities to inhibit DENV-2 replication in cell culture. These compounds show high efficacy in blocking DENV-2 replication, and dasatinib showed the inhibition of DENV-2 secretion. The exact mechanisms of these compounds have to be identified, but it is mediated by NS4B inhibition [165,166].

### 5.5. NS5 Targeting

NS5 is the largest flavivirus protein harboring RdRp as well as methyltransferase (MTase). It is therefore the most studied potential antiviral against flaviviruses [213]. The targeting of NS5 proteins in HCV has greatly affected in the treatment and management of HCV infection, resulting in a large array of studies aiming to identify specific NS5 inhibitors.

Antiviral compounds targeting NS5 can be classified into different categories, such as nucleoside inhibitors, non-nucleoside inhibitors, and MTase inhibitors.

Several nucleoside analogs have already been approved as antiviral drugs against herpesvirus [214], HCV [215,216], and HIV infection [217,218,219]. Through the screening and repurposing of drugs, various nucleosides analogs that target viral polymerases have been identified that have *in vitro* and *in vivo* activity against flaviviruses [220].

#### 5.5.1. Nucleoside Analogs

Several nucleoside analogs inhibitors have been investigated, and some are currently undergoing trials. Their antiviral action is related to the premature termination of RNA genome synthesis during replication.

BCX4430 (galidesivir), an adenosine analog, exhibited *in vitro* and *in vivo* antiviral efficacy against WNV, TBEV, and ZIKV via the inhibition of viral RNA polymerase through non-obligate RNA chain termination [167,168]. This drug is undergoing phase 1 trials, but no results are yet available.

Favipiravir, a nucleoside analog, is already licensed for use against the influenza virus and is undergoing clinical trials against the Ebola virus [221] and SARS-CoV-2 [222]. It can protect mice against WNV and YFV [170]. Favipiravir was also effective against ZIKV *in vitro*, leading to an increase in the number of mutations and promoting the production of defective viral particles [223]. In Cynomologus macaques, favipiravir treatment led to a statistically significant reduction in the plasma ZIKV viral load [171].

Balapiravir is a prodrug of a cytidine analog (R1479) that has *in vitro* and *in vivo* antiviral activity against HCV. A phase 1 trial against DENV infection [224] did not confirm these results, likely due to late treatment, a limited number of patients, suboptimal dosage, and a lack of *in vivo* phosphorylation of the drug. [172]. In a completed phase 1 trial, balapiravir was well tolerated, but the viral load was barely affected by treatment.

Another nucleoside analog, NITD-008, was tested both *in vitro* and *in vivo* for the treatment of ZIKV, TBEV, and DENV, yielding a significant viremia reduction [173,174]; however, it did not advance to clinical trials due to preclinical toxicity, such as weight loss, decreased motor activity, retching, feces with mucoid or blood, irreversible corneal opacities, blood abnormalities, and movement disorder [225].

A nucleoside analog that completed its phase 1 clinical trial (NCT04722627) and is now in phase 2 is AT-752, a prodrug of a guanosine nucleotide analog, which shows strong anti-viral *in vitro* activity against DENV-2, DENV-3, WNV, YFV, ZIKV, and JEV *in vivo* [175,176] as well as in an AG129 mouse model of DENV-2 and YFV, in which the viral load was significantly decreased, and the survival of AT-752-treated mice was clearly improved [175,176]. No results have been released regarding the phase 2 trial. Notably, AT9010, the active triphosphate metabolite of AT-752, was detectable in the peripheral blood mononuclear cells (PBMCs) of different animal models (rats, mice, and non-human primates) at a consistent level, supporting the use of AT-752. The viral activity of this molecule is currently under evaluation in a phase 1 trial.

The adenosine analog 7DMA (7-deaza-2′-C-methyladenosine), also called MK-608, was originally developed for HCV therapy. The use of 7DMA in HCV treatment failed due to negative but undisclosed results in a phase 2 clinical trial; it may have been due to mitochondrial toxicity [226]. It was subsequently tested for flavivirus treatment, and studies have observed antiviral effects on TBEV, ZIKV, WNV, and DENV *in vivo* [166,177,178]. Notably, ZIKV expression was strongly inhibited in Vero cells. In an AG129 mouse model, 7DMA treatment reduced the ZIKV viral load and halted disease progression. On the other hand, a study on BALB/c mice infected with WNV confirmed the efficacy of 7DMA in downregulating viremia; however, to maintain its antiviral efficacy, treatment must be administered in the early days of infection.

The strong structural homology among different RdRp enzymes belonging to different *Flaviviridae* is a pivotal characteristic that might be used for the possible development of a pan-*Flaviviridae* drug [227]. NS5 amino acid residues were predicted to interact with sofosbuvir and to show approximately 80% conservation among WNV, DENV, and ZIKV [228,229], suggesting the possible use of sofosbuvir in flavivirus therapy. Sofosbuvir represents a classic molecule successfully used for HCV treatment and is a uridine nucleotide analog pro-drug that is transformed in hepatocytes in its active form 2′-deoxy-2′-α-fluoro-β-*C*-methyluridine-5′-triphosphate. *In vitro* and *in vivo* experiments demonstrated antiviral action against ZIKV in different cell models [180,181]. De Freitas and colleagues [179] demonstrated that sofosbuvir inhibits *in vitro* YFV replication and protects YFV-infected mice, both neonatal and adult, from mortality and weight loss. Other drugs demonstrated a good activity *in vitro,* but *in vivo* or trial procedures treatment were not effective. Ribavirin is a synthetic nucleoside analog with a range of antiviral applications; it is particularly used for HCV and HBV treatment. Ribavirin does not suppress ZIKV replication in mice, produced no viral load decrease, and did not increase survival in mice infected with DENV or ZIKV [186].

#### 5.5.2. Non-Nucleoside Inhibitors

Several drugs form a complex with NS5, thereby inhibiting its enzymatic function. Among these, NITD-434 and NITD-640 target the RNA tunnel of RNA polymerase and display *in vitro* pan-flavivirus activity [182]. NITD-29, through binding with the NS5 N-pocket, was effective against all DENV serotypes [183].

Stefanik and coworkers [184] assayed a library of FDA-approved antiviral drugs for the ability to block flavivirus replication *in vitro*. Efavirenz (an inhibitor of the HIV-1 reverse transcriptase enzyme), tipranavir (a nonpeptidic HIV protease inhibitor that targets the HIV protease), and dasabuvir (an NS5B polymerase inhibitor that terminates the RNA polymerization of HCV) can inhibit WNV, ZIKV, and TBEV replication, suggesting a possible new application for these drugs in flavivirus treatment.

*In vitro* and *in vivo* treatment with lycorine, a benzyl phenethylamine alkaloid, resulted in reduced viral ZIKV replication in infected cells, and in the CNS, liver, and serum, as well as a downregulation in the inflammatory response in infected mice [190]. The cellular thermal assay demonstrated direct binding between lycorine and NS5, and further *in vitro* evaluation indicated that this binding inhibits RdRp activity. Lycorine can bind to HSP70 and NS3 and these interactions might play a role in ZIKV inhibition.

Ivermectin, a drug used extensively for parasite treatment, has shown antiviral activity against DENV and ZIKV in cell models through inhibiting IMP-α/β1, thus altering viral protein trafficking and NS5 trafficking [187]. A phase 2/3 clinical trial demonstrated an accelerated NS1 clearance in DENV patients but no clinical efficacy at the chosen dosage [188].

In contrast, emetine, a drug approved for amoebiasis treatment, demonstrated consistent antiviral activity [189] in cell cultures and mouse models infected with ZIKV, wherein viral replication was inhibited. Three interesting features were observed: i) emetine directly binds to NS5, as shown in an cellular thermal assay and molecular docking experiments; ii) emetine treatment induces a significant decrease in NS1 protein levels; and iii) emetine accumulates in cellular lysosome with a derangement of lysosomal function and impaired autophagy, thus interfering with cellular trafficking and the regulation of viral infection [191].

Dolutegravir is a broad-spectrum matrix metalloproteinases inhibitor [230] that was tested for its potential use against flavivirus. It is mainly known for its high efficacy in suppressing HIV replication in deintensification and monotherapy [231,232], for HIV patients with a small reservoir [233] with minor side effects. Due to its high tolerability, it is a promising molecule for flaviviral therapy. Experiments with 12 FDA-approved drugs revealed that dolutegravir does not reduce flaviviral replication but that it effectively inhibited ZIKV-mediate cytopathic effects with >90% viability of infected Vero cells [184].

Molecular docking analysis suggested that compound TPB binds to the catalytic active site of RdRp and likely blocks viral RNA synthesis with an allosteric effect. *In vitro* and *in vivo* studies of this compound resulted in significantly reduced ZIKV viremia [192]. AR-12 a celecoxib-derivative cellular kinase inhibitor with a broad spectrum of antiviral activities, downregulates the PI 3-kinase/Akt (PKB) pathway, glucose-regulated protein 78 (GRP78), and dihydroorotate dehydrogenase (DHODH) in DENV-infected cells. AR-12 treatment in mice determines the derangement of non-structural protein expression and the subsequent production of new viral progeny. Two derivatives of AR-12, P12-23 and P12-34, can exert antiviral effects on DENV, ZIKV, and JEV. These compounds block pyrimidine biosynthesis de novo, inducing the failure of viral replication process [185].

#### 5.5.3. MTase Inhibitors

Several MTase inhibitors have been investigated. Among these, sinefungin, an S-adenosyl-L-methionine (SAM) analog, is a broad-spectrum inhibitor of DENV and WNV. Sinefungin competes with SAM to bind with the SAM site on NS5 viral proteins. Due to its substitution of methylated sulfur with amine and carbon, it can bind to the SAM site but without completing all the interactions with NS5. Despite this, sinefungin binds viral NS5 with six-times greater affinity [193,194,195].

Non-structural proteins are excellent candidates for therapeutic targets since they are essential for viral replication and frequently have conserved structures. Moreover, the idea of a pan-dengue and perhaps a pan-flavivirus antiviral is conceivable due to the structural similarity between essential NS proteins. With the remarkable similarity of flaviviruses and its significance in viral replication, NS4B in particular appears to be an attractive target. Additionally, it has been demonstrated that pharmacologically inhibiting the NS3 protease and the RdRp NS5 is effective in halting viral replication. The NS proteins are well-conserved; therefore, drugs that bind to them typically have less activity altered by resistance mutations. This is a significant benefit of targeting the NS proteins.

## 6. Assembly and Egress Inhibitors

Flavivirus virion assembly takes place in membranous structures associated with the ER, where E, C, and prM heterodimers associate. Viral particles are eventually transported through secretory pathways to the Golgi apparatus, where the maturation and N-linked glycosylation of prM and E proteins takes place. Here, a decrease in pH causes a conformational shift in prM-E spikes, and in this acidified compartment, cellular host protease furin cleaves prM, maturing the virion that is subsequently released from the cell by vesicular fusion with the plasma membrane [234]. As many cellular host compartments and proteins are involved in virion assembly and release, targeting these may be an effective antiviral strategy (Table 4).

The glycosylation of viral proteins is an important mechanism that takes place in the secretory pathway, and Flaviviruses strongly rely on the prM, E, and NS1 glycosylation for infectivity. The derangement of viral glycoproteins glycosylation during virion assembly alters the last steps of the viral cycle. Several studies have focused on the potential antiviral activity of ER α-glucosidase I and II enzyme inhibitors. Iminosugars are a class of molecules that can inhibit these enzymes and induce a derangement of viral glycosylation associated with viral structural glycoprotein misfolding. Two ER α-glucosidase I and II inhibitors were assayed: celgosivir and UV-4B. Celgosivir is an oral iminosugar prodrug of castanospermine that impedes the processing of the N-linked oligosaccharides of viral envelope glycoproteins and NS1 by preventing the removal of the terminal glucose residue from N-linked glycans [235]. The defective processing of N-linked oligosaccharides of viral envelope glycoproteins elicits a derangement of virion structure, particularly of prM and E viral glycoproteins, along with the inhibition of mature virion formation [236]. Several investigations have shown its antiviral effect against DENV infection through *in vitro* models and AG129 mice. Celgosivir treatment induces significant viremia reduction only when administered at the beginning of the infection. Two clinical trials were conducted to evaluate celgosivir in terms of pharmacokinetics, activity, safety, and tolerability in patients with DENV, but viral load was not reduced, and the trial was dismissed [237,238,239].

UV-4B reduces the infectious virus titer and RNA of all four DENV serotypes in cell culture. In a lethal ADE DENV-2 mouse model, UV-4B protected against lethal DENV infection even when treatment started 48 h post-infection. A phase 1 trial was completed, in which this molecule did not elicit serious adverse events [240].

Other iminosugars and their derivatives have exhibited antiviral effects on DENV infection, such as UV-12, CM-9-78, and CM-10-18, suggesting that this class of compounds might represent a promising field in antiviral studies [241,242,243].

PF-05175157, TOFA (5-tetradecyloxy-2-furoic acid), and MEDICA 16 (3,3,14,14-tetramethylhexadecanedioic acid) are acetyl-Coenzyme A carboxylase (ACC) 1 and 2 inhibitors that are involved in host metabolism regulation. Studies on their use in flavivirus treatment demonstrated that viral progeny exhibited an incomplete morphogenesis. Antiviral effects were detected against different flaviviruses, including WNV, DENV, and ZIKV *in vitro*. Analysis of their antiviral effects in mouse models infected with WNV demonstrated a reduction in viral load, but subsequent experiments involving ACC-2-negative mice indicated that the inhibition of both ACC1 and ACC2 is required for a full viral-inhibiting effect [244,245].

SFV785 has selective effects on MAPKAPK5 kinase activity and has been inhibiting DENV and YFV viral yield by altering the recruitment and assembly of nucleocapsid during DENV assembly, thus reducing the production of infectious virions [246,247].

Lovastatine, an HMG-CoA reductase inhibitor, is a statin drug, which was proposed as an anti DENV compound for its reducing action on the synthesis of cholesterol and isoprenoid, and the alteration of glycosylation. The hypothesis was based on the essential presence of glycosylated proteins on the cell membrane surface for viral entry and for the lipid bilayer needed for the assembly and release of infectious viral particles. Unfortunately, it did not show any antiviral activity *in vivo* and in clinical trials [248,249,250].

A recent paper demonstrated a significant reduction in infected cells through a combination of two anti-cholesterol drugs: atorvastatin and ezetimibe. These drugs acted synergistically in the reduction of DENV-2 infection, while their effect was only additive for when concerning DENV-4 and ZIKV, and antagonistic in YFV-infected cells [251].

Both capsid and glycoprotein assembly are meticulously planned processes that were related to activation by both known and unknown effectors. Reduced virus propagation can be obtained, interfering the kinetics of viral assembly and the release by infected cells and viral maturation.

**Table 4 microorganisms-11-02427-t004:** Flavivirus assembly and egress inhibitors.

Target	Drug	Viral Specificity	Study Stage	Ref.
Assembly	Celgosivir	DENV	phase 1/2	[237,238,239]
	UV-4B	DENV	phase 1/2	[240]
	UV-12	DENV	*in vivo*	[241,242,243]
	CM-9-78			
	CM-10-18			
	PF-05175157	WNV, DENV, ZIKA	*in vivo*	[244]
	TOFA		*in vitro*	[244,245]
	MEDICA 16		*in vitro*	[244,245]
	SFV785	DENV	*in vitro*	[246,247]
		YFV		
	Lovastatine	DENV	*in vivo*	[248,249,250]
	Atorvastatine, Ezetimibe	DENV-2,-4; ZIKA	*in vitro*	[251]
C	VGTI-A3	DENV-2	*in vitro*	[245]
	VGTI-A3-03		*in vitro*	[252]
	ST-148		*in vivo*	[253,254]

### C Protein Targeting

C protein, which has a molecular weight of 11 kDa, may pack viral RNA and is the fundamental component of nucleocapsids. The C protein dimer has four helix structures in each of its monomer molecules. Its interaction with viral RNA depends on C protein dimerization. The capsid protein, although it is the least conserved among the flaviviruses, is very interesting due to its multiple functions and its great structural flexibility. It is precisely for this reason that it is the focus of numerous new studies as a new therapeutic target for infections caused by flaviviruses.

Currently, there are three molecules known for their antiviral activity, which have the binding with the viral capsid protein as a mechanism of action: VGTI-A3, VGTI-A3-03, and ST-148.

Smith and collaborators [245] found that VGTI-A3 (PubChem ID: 4259739) is a small chemical with strong virus specificity and significant antiviral activity that is able to prevent DENV serotype 2 viral multiplication. Going deeper in their analysis of selected compounds and by a structural-activity relationship (SAR) analysis, Smith collaborators found that a VGTI-A3 analog, called VGTI-A3-03, showed higher antiviral activity and greater solubility, maintaining DENV-2 specificity. In particular, VGTI-A3-03 acts by binding and incorporating C protein into the DENV virions and reducing the *in vitro* infectivity of the released DENV-2 virion particles. The mechanism of action seems to be the compound mediated stabilization of capsid dimer–dimer interactions, thus impeding disassembly after entry [252].

Another molecule, ST-148, similarly causes C protein to undergo tetramerization, which is incorporated into progeny virions that are unable to be properly uncoated [253,254].

The C protein has many more uses than the structural role in virion structure. It can interact with several host proteins to promote virus multiplication in addition to being responsible for encapsidation to protect the viral RNA. Consequently, the C protein is crucial for the viral life cycle and infected host cells [255].

## 7. Unknown Target

Some tested drugs with antiviral effects against *Flaviviridae* did not show a clear action mechanism of action (Table 5). For example, nitazoxanide, amodiaquine, lanatoside C, bromocriptine, hippeastrine hydrobromide (HH), and azithromycin exert antiviral effects against one or more flaviviruses *in vitro* and *in vivo* [256,257,258,259]. In some cases, as for lanatoside C, the narrow therapeutical index dose not elicit the use of these drugs. In addition, bromocriptine, an agonist of dopamine receptors D2 and D3, showed an effect *in vitro* but not in mouse models.

In contrast, HH showed antiviral activity against avian influenza (H5N1) and HCV, but subsequent studies have demonstrated its ability to suppress RNA replication and the formation of infectious particles in ZIKV-infected human neural progenitor cells (hNPCs). In human fetal-like forebrain organoid cultures, HH was effective at infection control, and in a mouse model, ZIKV infection was effectively inhibited with a significant decrease in ZIKV RNA in the brain and ZIKV-induced cellular apoptosis.

Moreover, amodiaquine, an antimalarial drug, suppressed ZIKV infection in hNPCs and an SCID-beige mouse model [260] whereas azithromycin, a macrolide, reduced the viral load in ICR mice, although no data about its mechanism are available.

## 8. Artificial microRNAs

Another strategy under investigation is artificial microRNAs (amiRNAs) specifically directed against viral genomes and the antiviral effect of human miRNAs, that is, the reduction of neurovirulence and viral infection of TBEV, DENV, and JEV [261,262,263].

Vaccine preparations against flaviviruses are based on live attenuated or inactivated viruses, but their efficacy is moderate; they cause severe side effects and lead to the appearance of reverted variants. In recent years, a new type of compound, based on genetically modified viruses, is helping to improve efficacy and side effects. The underlying principle is the insertion of microRNA recognition elements (MREs) in a specific position on the flavivirus genomes, 3′UTR, which is highly conserved throughout flaviviruses, and it plays an essential role in viral replication and translation stages, as well as virulence [264,265,266,267].

This new type of vaccines can induce neutralizing antibody production in mice [268] and could be a promising strategy to avoid the antibody-dependent enhancement (ADE) phenomenon, which causes increased viral replication due to the production of specific antibodies against viruses that facilitates viral entry, upregulates autophagy, and inhibits interferon signaling. For some viruses, the ADE phenomenon is not abolished with MRE-based vaccines, but this question could be overcome by proposing miRNAs not as a viral genome modification, but as an active compound of a drug used as prevention or therapeutic treatment, as shown in a recent review [269].

## 9. Trials

Considering that a higher viral burden could promote severe disease, the identification of flavivirus antivirals has been an important focus of research for therapeutics, and multiple clinical trials have been conducted on antivirals cited in this review. Clinical trials with repurposed drugs, such as balapiravir, chloroquine, lovastatin, and celgosivir, having antiviral activity in preclinical investigations have not yet demonstrated any effectiveness in lowering viremia or positive clinical outcomes.

Several trials investigating new drugs for flavivirus treatment are underway or completed (Table 6). According to https://clinicaltrials.gov (accessed on 16 August 2023), 17 compounds are under investigation (Table 6), both direct-acting and host-factor-targeting antivirals acting on different viral life cycle steps. Six studies are being performed on drugs that act on virion entry and the fusion of virions, two on drugs targeting NS4B, and three on molecules that inhibit NS5 polymerase activity. Furthermore, six additional trials are assessing host-directed antivirals. These molecules are being evaluated for their antiviral activity against DENV, ZIKV, YFV, and JEV.

Some studies have been discontinued for various reasons, including a lack of antiviral activity, patients (for ZIKV), or funding, as described in Table 6 (see the “Note” column). All these studies were at phase 1 or 2, and only ivermectin is in phase 3, but no results are yet available despite the negative results published by Caly and coworkers [153].

Another possible explanation of the premature termination of clinical trials is related to the inefficacy of some of these molecules. In addition, the SARS-CoV-2 pandemic has catalyzed media attention and funds allocated to other research topics. Given that at this moment the epidemic is regressed and under control, we can hope that the pharmaceutical industries can reprogram their lines of research and orient their priorities by returning to research on antivirals directed against flaviviruses.

## 10. Conclusions

The absence of approved antivirals against flavivirus led to numerous *in vitro* and *in vivo* studies. These are focused on specific molecules that can interfere with one or more steps of the viral cycle. Molecules with antiviral activity can exert a direct action on various viral molecules, in particular E glycoprotein, and the non-structural proteins NS3 and NS5 or on non-viral targets, which play an important role in the correct development of the viral replication cycle, even though they can have consistent side effects. The homology of some regions of the viral proteins among the various flaviviruses can be used to find drugs with effective antiviral activity against different flaviviruses. Numerous molecules have been proposed, including synthetic peptides and putative antiviral molecules revealed through in silico studies [125,279,280]. However, several compounds currently used against other targets (such as bacteria and helminths) have been successfully tested *in vitro* for use on flaviviruses; however, some have not yet been tested *in vivo.*

Many antiviral compounds have not yet progressed beyond testing in cell lines and mouse models, while others have begun testing in trials. Most of these trials are ongoing and almost all of them are in phase 1 or 2. Despite this, many studies have not confirmed *in vitro* results or have been discontinued. Perhaps one of the reasons for the current lack of viable anti-flavivirus drug is a limited interest by sponsors, given that most cases of not HCV flaviviral infections (with the notable exception of WNV) are in low-income countries [8]. The targets of antiviral treatments against flaviviruses are related to different steps of the viral cycle, such as halting viral entry into the target cell, inhibiting viral replication, and preventing the severe damage that is sometimes detected after the peak of viraemia. The possibility of counteracting different stages of the viral cycle indicates that an antiviral strategy based on a cocktail of antivirals [281,282,283], acting on different steps of the replicative cycle, may be useful for overcoming or reducing the importance of genetic mutations; such a drug combination must be carefully evaluated, and a useful synergy must be found for increased treatment efficacy.

Multidrug therapy is a well-known procedure employed in HIV, HBV, and HCV treatments. In particular, these combinations are very effective at tackling viral replication in different stages, decreasing the impact of antiviral resistance. Multidrug combination is a promising strategy that could be also used for flaviviruses treatment. In fact, the use of different drugs acting on different viral targets reduces the risk of drug resistance.

Despite the large variety of viral targets studied for different flaviviruses, the complete potential of many molecules has yet to be unrevealed. It is vital to continue developing and implementing countermeasures that restrict flavivirus transmission and disease. Surveillance programs are essential to investigate and control pathogen spread and geographical localization by public health authorities. This review aims to outline the importance of developing new drugs for the treatment of these viruses that are increasing in the world. Until now, most flavivirus-endemic countries are located in tropical and sub-tropical areas and are not equipped with hospitals and trained personnel. However, climate change is going to shift this reality [284], and the lack of viral drugs will be an issue even for the richest countries.

## Figures and Tables

**Figure 1 microorganisms-11-02427-f001:**
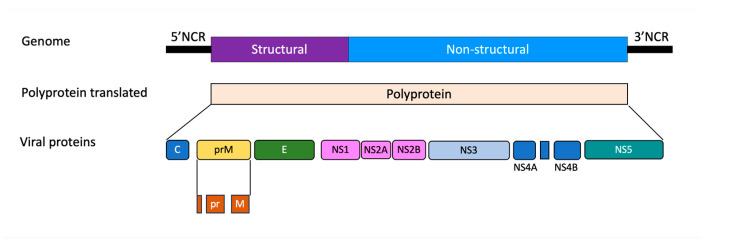
Flaviviral genome. Flaviviral genome contains genes for structural and non-structural proteins, which are flanked by two non-coding regions (NCR) at 5′ and 3′. Encoded Polyproteins are cleaved by cellular and viral proteases, releasing three structural proteins and seven non-structural ones. Protein M is initially released as a precursor (prM) and subsequently cleaved and released as a mature protein M. Modified by King and colleagues [14].

**Figure 2 microorganisms-11-02427-f002:**
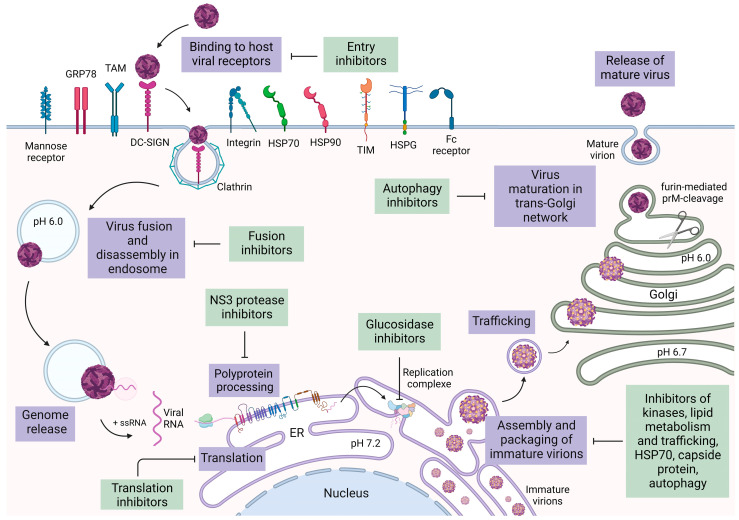
Flavivirus life cycle. The viruses bind to receptors on the host cell and enter via receptor-mediated endocytosis. The viral envelope merges with the host membrane in endosomes, and the viral capsid disassembles, allowing the viral genome to enter the cytoplasm. The positive-sense RNA is translated in the ER into a single polyprotein, which is co- and post-translationally digested by viral and host proteases. In specialized ER-derived membrane compartments, the viral-RNA-dependent RNA polymerase replicates the viral genome. The assembled viral nucleocapsids sprout into the ER lumen and exit the cell via the secretory route. Non-infectious, immature viral and subviral particles are produced and transmitted by the trans-Golgi network. The host protease furin cleaves the immature virion particles, resulting in mature, infectious particles, which are subsequently released by exocytosis [78,79].

**Figure 3 microorganisms-11-02427-f003:**
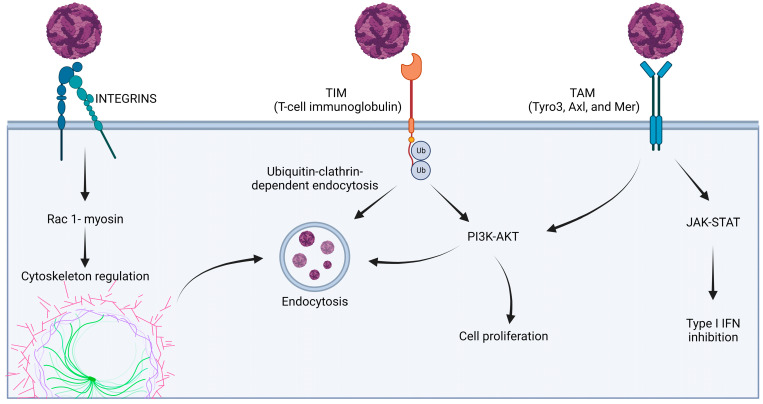
Signaling mechanisms of viral entry receptors. Flavivirus entry receptors affect a variety of pathways, including cytoskeleton alterations via integrins, endocytosis, proliferation, and cell survival via PI3K-AKT (TIM and TAM receptors), and immune response downregulation by JAK-STAT.

**Table 1 microorganisms-11-02427-t001:** Cell membrane receptors that interact with viral E glycoprotein are involved with E protein binding and consequent viral entry or specifically recognize each cell receptor.

Host Receptor	Virus	Ref.
α_v_β_3_integrin	JEV	[21,22]
	WNV	[23]
α_v_β_5_integrin	ZIKV	[24]
C-type leptin receptors	DENV	[25]
	JEV	[26]
	WNV	[27,28,29]
TIM-1 and TAM receptors	DENV	[30]
	WNV	[31]
	YFV	[31]
	ZIKV	[32,33]
DC-SIGN-mediated	JEV	[34]
	ZIKV	[32]
	DENV	[35]
Chondroitin sulphate E	DENV	[36]
Neolactotetraosylceramide (nLc4Cer)	DENV	[37]
Heparan sulphates (HSs)	DENV	[36]
Heparan sulphates proteoglycans (HSPGs)	DENV	[36,38]
Sphingomyelin	JEV	[39]
Mannose receptor	DENV	[40]
CLEC5A	DENV	[40]
	WNV	[41]
Claudin-1	DENV	[42]
	WNV	[43]
Heat Shock Cognate Protein 70 (HSCP70)	DENV	[44,45,46]
	JEV	[47,48]
Heat Shock Cognate Protein 90 (HSCP90)	DENV-2	[45,49]
Heat Shock Cognate Protein 90-β (HSCP90β)	JEV	[50]
Glucose-regulated protein (GRP78)	DENV-2	[51]
	JEV	[52,53]
Vimentin	JEV	[54,55]
CD14	JEV	[56]
37/67 kDa high-affinity laminin receptor	JEV	[56]
	DENV-1	[57]
Nucleolin	JEV	[56]
Dopamine D2 receptor	DENV	[58]
	JEV	[59]
Dopamine D4 receptor	DENV	[60]
Heat Shock Protein 70 (HSP70)	DENV-2	[46,61]
	JEV	[47]
	ZIKV	[62]
NKp44	WNV	[63]
CD300a	DENV-4	[64]
	YFV	[64]
α2,3-linked sialic	ZIKV	[65]
Prohibitin ½	DENV-3	[66]
PLVAP	JEV	[67]
GKN3	JEV	[67]
GAGs	DENV	[68]
	JEV	[68]
	TBEV	[68]
	WNV	[68]
	YFV	[68]
NCAM1	ZIKV	[69]
PtdSer receptor	DENV	[64]

**Table 3 microorganisms-11-02427-t003:** Flavivirus viral replication, polyprotein synthesis, and processing inhibitors.

Target	Drug	Viral Specificity	Study Stage	Ref.
NS1	Peptide 3, 4, 10, 11	DENV	*In vitro*	[125,126]
	Deoxycalyxin-A	ZIKV	*In silico*	[127]
	mAb AA12	ZIKV	*In vivo*	[128]
	mAb 2B7	DENV, WNV, ZIKV	*In vitro*	[129]
	Castanospermine and Deoxynojirimycin	DENV, ZIKV	*In vitro*	[130]
	N-nonyl-Deoxynojirimycin	DENV-2, JEV	*In vitro*	[131]
	Celgosivir	DENV	Phase 1 clinical trial	[132,133]
NS2A	shRNA and siRNA	JEV	*In vitro*	[134,135]
NS2B-NS3	Novobiocin	DENV, ZIKV	*In vivo*	[136,137]
	Temoporfin	DENV, YFV, WNV, JEV	*In vitro*	[114]
		ZIKV	*In vivo*	[114]
	JMX0207	DENV-2	*In vitro*	[138]
		ZIKV	*In vivo*	[138]
	Nelfinavir	DENV-2	*In vitro*	[139]
	Compound 4	ZIKV	*In vitro*	[140]
	Compound 14, Compound 15	DENV	*In vitro*	[141]
	NSC135618	DENV, ZIKV, WNV, YFV	*In vitro*	[142]
	ZP10	ZIKV	*In vitro*	[143]
	Aprotinin	DENV	*In vitro*	[144,145]
		WNV	*In silico*	
	Hydroxychloroquine	ZIKV	*In vivo*	[146,147]
	Methylene blue	ZIKV	*In vivo*	[148]
		DENV	*In vitro*	[138,149,150]
	Erythrosin B	ZIKV, DENV	*In vitro*	[138,149,150]
	Ivermectin	YFV, WNV	*In vitro*	[151]
		ZIKV	*In vivo*	[152]
		DENV	Phase 2/3 clinical trial	[153]
	Bortezomib	ZIKV, DENV	*In vitro*	[154]
	Asunaprevir, Simeprevir	ZIKV	*In vitro*	[155]
NS3	ST-610	DENV	*In vivo*	[156]
	Suramin	DENV, ZIKV	*In vitro*	[157]
NS2B	NSC157058	WNV	*In silico*	[158]
		ZIKV	*In vivo*	
NS4A	Compound B and SBI-0090799	DENV, ZIKV	*In vitro*	[159,160]
NS3-NS4B	JNJ-A07	DENV	*In vivo*	[161]
	JNJ-64281802	DENV	Phase 2 clinical trial	[161]
	JNJ-1802	DENV	Phase 1 clinical trial	[162]
		JEV, WNV, ZIKV	*In vitro*	[162]
NS4B	NITD-688	DENV	*In vivo*	[163]
	Manidipine	JEV	*In vivo*	[164]
		DENV, ZIKV, WNV	*In vitro*	
	AZD0530, Dasatinib	DENV-2	*In vitro*	[165,166]
NS5	Galidesivir	WNV, TBEV, ZIKV	*In vivo*	[167,168]
		YFV	Phase I clinical trial	[169]
	Favipiravir	WNV, YFV	*In vitro*	[170]
		ZIKV	*In vivo*	[171]
	Balapiravir	DENV	Phase 1/2 clinical trial	[172]
	NITD-008	ZIKV, TBEV, DENV	*In vivo*	[173,174]
	AT-752	DENV, YFV	Phase II clinical trial	[175,176]
		WNV, ZIKV, JEV	*In vitro*	[175,176]
	7DMA	TBEV, ZIKV, WNV, DENV	*In vivo*	[166,177,178]
	Sofosbuvir	YFV	*In vivo*	[179]
		ZIKV	*In vivo*	[180,181]
	NITD-434, NITD-640	Pan-flavivirus	*In vitro*	[182]
	NITD-29	DENV	*In vitro*	[183]
	Efavirenz, Tipranavir, Dasabuvir	WNV, ZIKV, TBEV	*In vitro*	[184]
	AR-12	DENV	*In vivo*	[185]
	P12-23, P12-34	DENV, ZIKV, JEV	*In vitro*	[185]
	Ribavirin	ZIKV	*In vivo*	[186]
	Ivermectin	ZIKV	*In vitro*	[187]
		DENV	Phase 2/3 clinical trial	[188]
	Emetine	ZIKV	*In vivo*	[189]
	Lycorine	ZIKV	*In vivo*	[190]
	Dolutegravir	ZIKV	*In vitro*	[191]
	Compound TPB	ZIKV	*In vivo*	[192]
	Sinefungin	WNV, DENV	*In vivo*	[193,194,195]

**Table 5 microorganisms-11-02427-t005:** Flavivirus antivirals with unknown mechanisms.

Drug	Target	Viral Specificity	Study Stage	Ref.
Nitazoxanide	Unknown	ZIKV	*In vivo*	[256,257,258,259]
Amodiaquine				
Lanatoside C				
Bromocriptine				
Hippeastrine hydrobromide				
Azithromycin				

**Table 6 microorganisms-11-02427-t006:** Drugs against flaviviruses undergoing clinical trials. List of drugs which are under investigation, with information regarding target, viral specificity, mechanism of action, clinical trial ID, stage of clinical trial and relative status, available results, and references.

Antiviral	Target	Virus	Mechanisms	Clinical Trial Identifier	Clinical Trial	Status	Note	Ref.
**DIRECT-ACTING ANTIVIRALS**								
Dengushield	E protein	DENV	Neutralizing and fusion-inhibitory activity	NCT03883620	Phase 1	Completed	No results available	[270]
TY014	E protein	YFV	Neutralizing and fusion-inhibitory activity	NCT03776786	Phase 1	Completed	Safe and reduces symptoms of YFV vaccines	[271]
ZIKV-Ig	E protein	ZIKV	Neutralizing and fusion-inhibitory activity	NCT03624946	Phase 1	Completed	ZIKV-Ig was safe and well tolerated	[100]
Tyzivumab	E protein	ZIKV	Neutralizing and fusion-inhibitory activity	NCT03443830, NCT03776695	Phase 1/2	NCT03443830: completed; NCT03776695: withdrawn	NCT03443830: no results available, NCT03776695 withdrawn due to lack of Zika infected patients	[272,273]
MGAWN1	E protein	WNV	Neutralizing and fusion-inhibitory activity	NCT00515385, NCT00927953	Phase 1/2	Withdrawn due to low enrollment	Safe and well tolerated in healthy subjects	[101]
IVIG	E protein	JEV	Neutralizing and fusion-inhibitory activity	NCT01856205	Phase 2	Completed	Development of neutralizing antibodies in JEV positive patients.	[274]
JNJ-1802	NS4B	DENV	Induction of conformational changes	-	Phase 1	Completed	Good safety and pharmacokinetics	[211,212]
JNJ-64281802	NS4B	DENV	Induction of conformational changes	NCT05201794, NCT04906980	Phase 2	Ongoing	Dengue prophylaxis in healthy individuals (NCT05201794) and dengue therapy in patients with confirmed dengue fever (NCT04906980)	[161]
Balapiravir	NS5	DENV	Inhibit RdRp	NCT01096576	Phase 1	Completed	Discontinued: well tolerated but did not reduce viremia nor fever clearance time	[224]
Galidesivir	NS5	YFV	Inhibit RdRp	NCT03891420	Phase 1	Ongoing	-	[169]
AT-752	NS5	DENV, YFV	Inhibit RdRp	NCT04722627, NCT05466240, NCT05366439	Phase 2	Ongoing	-	[175,176]
**HOST-DIRECTED ANTIVIRALS**								
UV-4B	ER α-glucosidase I and II	DENV	Function inhibition leading to defective processing of N-linked oligosaccharides of viral envelope glycoproteins	NCT02061358	Phase 1	Completed	No serious adverse events reported	[240]
Chloroquine	Endosomal acidification	DENV, ZIKV	Alkalinization of intracellular organelles acidification	NCT00849602	Phase 1/2	Completed	CQ does not reduce the durations of viraemia in dengue patients	[107,113]
Celgosovir	ER α-glucosidase I and II	DENV	Defective processing of N-linked oligosaccharides of viral envelope glycoproteins	NCT01619969, NCT02569827	Phase 1/2	Completed	NCT01619969: several non-significant trends of pharmacological effect of Celgosivir. NCT02569827: withdrawn due to lack of funding	[237,239]
Metformin	AMPK	YFV	Reduction in lipid synthesis by activating AMP-activated protein kinase (AMPK)	NCT04267809	Phase 2	Ongoing	-	[275]
IC-14	CD14	DENV	CD14 antagonist antibody	NCT03875560	Phase 2	Withdrawn	Seeking funding	Patent: WO2018165720A1 [276]
Ivermectin	IMPα/β (Host)	DENV	Inhibition of the IMP α/β -mediated nuclear import	NCT02045069, NCT03432442	Phase 2/3	Completed	No results available	[277,278]

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
