# Peer review of "Vector-Transmitted Flaviviruses: An Antiviral Molecules Overview"

_microorganisms, 2023, doi:10.3390/microorganisms11102427_

Round 1
Reviewer 1 Report
The authors presented an intensive review of antiviral molecules and their target molecules or pathways during infection with Flaviviruses. The manuscript is very well written and relevant to the field. The only adjustment I recommend is to delete the accession number from Table 1. To keep the same conformation of the table, I would suggest replacing the accession number with the virus family.
Author Response
Thank you for taking the time to review this manuscript and your suggestions. Below we provide the point-by-point response to each of your requests.
Please see the attachment.

Reviewer 2 Report
The manuscript describes that research progress in the treatment of viruses belonging to the family Flaviviridae. It is particularly interesting for potential readers such as drug developers and healthcare workers. There are some aspects that need attention by the authors.
Comments to the Author:
1. In the section of conclusions, the author should provide their own views on the reasons behind the current unfavorable development of viral disease treatment drugs, as well as potential solutions.
It is crucial for the author to express their personal opinions and insights regarding the reasons for the current challenges faced in the field of viral disease treatment drugs. Additionally, offering possible solutions to address these issues is essential.
2. The table in the manuscript should be three line table. I don't think Table 1 is important for the entire manuscript, so I suggest deleting it.
3. English is not good and requires careful editing to improve grammar, tense, and consistency.
For example, in page 2, line 3, the sentence ’These epidemiological and clinical………’ is missing predicate.
4. In terms of organizing the article, further classification and introduction of therapeutic drugs based on different viruses would enhance readers' understanding and provide a valuable reference for this manuscript.
The English language of this article is generally readable, but minor editing of English language required.Author Response
Thank you for taking the time to review this manuscript and your suggestions. Below we provide the point-by-point response to each of your requests.
Please see the attachment.

Reviewer 3 Report
The manuscript provides a comprehensive overview of antiviral treatments against flaviviruses, covering a wide range of topics from molecular mechanisms to ongoing clinical trials. The paper is well-structured, with clear subsections that guide the reader through the complexities of the subject matter. However, there are areas where the paper could be improved for clarity and depth.
Strengths:
Comprehensive Coverage: The paper covers a wide array of antiviral strategies, from molecular inhibitors to host-directed therapies.
Up-to-date Information: The manuscript includes the latest clinical trials, making it a valuable resource for researchers in the field.
In-depth Analysis: The paper delves into the molecular mechanisms of action for various antiviral compounds, providing a nuanced understanding of the topic.
Areas for Improvement:
Clarity and Terminology: Some sections could benefit from clearer explanations, especially for readers who may not be experts in the field.
Discussion on Limitations: While the paper mentions ongoing clinical trials, it could benefit from a more in-depth discussion on the limitations of current antiviral strategies, including why some trials were discontinued.
Future Directions: The conclusion could be strengthened by including suggestions for future research directions, especially considering the rapid evolution of flaviviruses and the urgent need for effective treatments.
Data Presentation: While the manuscript is rich in content, the inclusion of more tables, figures, or flowcharts could enhance the reader's understanding of the complex mechanisms and numerous molecules discussed.
Referencing: Ensure that all cited work is relevant and up-to-date. Some references could be more directly tied to the statements they are supporting.
In summary, the paper is a valuable contribution to the field but would benefit from revisions for clarity, depth, and a more thorough discussion on the limitations and future directions of antiviral treatments against flaviviruses.
Comments by section:
Abstract
Major Comments:
Line 10-21: The abstract is generally well-structured but could benefit from further refinement to align with scholarly standards. Specifically, the abstract could be improved by adhering to the structured format: Background, Methods, Results, and Conclusions. This would provide a more comprehensive and objective representation of the article.
Minor Comments:
Line 17-18: The phrase "antiviral treatments are not available for these infections" could be misleading. It would be more accurate to state that "there are limited antiviral treatments," as some treatments are in clinical trials.
Title
Major Comments:
Line 2: The title "Flavivirus transmitted by vectors: an antiviral molecules overview" could be more precise. Consider revising it to "Vector-Transmitted Flaviviruses: A Comprehensive Review of Antiviral Molecules."
Minor Comments:
Line 2: The term "overview" might not capture the depth of the review. Consider using "Comprehensive Review" to indicate the scope of the paper.
Keywords
Major Comments:
Line 22: The keywords are relevant but could be organized better. Consider alphabetizing them or grouping them by theme for easier reference.
Minor Comments:
Line 22: Consider adding "public health" as a keyword, given the paper's focus on a significant public health issue.
General Comments
Line 5-9: The affiliations and correspondence are clearly stated, but consider adding ORCID iDs for all authors for better scholarly practice.
Line 14-16: The statement about the viruses' high morbidity and mortality rates is strong but would be strengthened by specific statistics and references.
Line 19-21: The abstract mentions ongoing studies but does not provide any references. Consider adding citations to substantiate this claim.
Line 16-18: The paper mentions altered climatic and social conditions as factors for the geographical range expansion of the viruses. This is a significant point that could be elaborated upon in the main text, supported by current literature.
Line 19-21: The abstract mentions challenges in clinical trials but does not elaborate. This could be a significant point to explore in the main text, perhaps in the Discussion section.
Introduction
Major Comments:
Line 26-27: The introduction begins with a statement about the high morbidity and mortality rates of flavivirus infections. While this is a strong opening, it would benefit from further contextualization. What is the global or regional impact of these rates? Are there specific populations that are more affected than others?
Line 35-38: The paper mentions the increasing geographical diffusion and incidence of flaviviruses due to various factors such as climate change and urbanization. This is a significant point that warrants further elaboration and should be supported by current literature.
Line 49-53: The discrepancy in the availability of antiviral drugs for Hepaciviruses and Flaviviruses is mentioned. This is an important point but could be further strengthened by discussing the economic or logistical challenges in developing antivirals for flaviviruses.
Line 54-56: The introduction concludes by stating the aim of the review. While this is clear, it could be more explicitly stated. For instance, what specific antiviral mechanisms will be discussed? Will the paper also address ongoing clinical trials or future directions in the field?
Minor Comments:
Line 28-31: The sentence structure here is somewhat convoluted. Consider revising for clarity. For instance, "The Flavivirus genus includes several viruses such as Dengue virus (DENV), Zika (ZIKV), West Nile (WNV), among others, which are well-known causative agents of human diseases."
Line 42-44: The statement "These Flavivirus caused up to 400 million of cases per year" could be more precise. Is this a global figure? Also, consider revising to "These Flaviviruses account for up to 400 million cases per year globally."
Line 46: The term "emergent health risk" could be more scholarly. Consider revising to "emerging public health concern."
Line 52-53: The sentence "from asymptomatic infections to severe disease" could be more specific. Are there particular flaviviruses that are more likely to result in severe disease?
Line 55-56: The mention of "Supplementary Table 1" is somewhat abrupt. Consider introducing it more smoothly, such as "For a comprehensive list of putative drugs with antiviral activity against flaviviruses, refer to Supplementary Table 1."
2. Biology of Flaviviruses
Major Comments:
Line 59-65: The section provides a detailed overview of the Flavivirus structure and genome. However, the relevance of this information to the paper's main focus on antiviral molecules is not explicitly stated. Consider adding a sentence that links this biological background to the challenges or opportunities in antiviral drug design.
Line 73-76: The paper mentions the varying degrees of homology among flavivirus genomes. While this is an important point, it would benefit from further elaboration. Specifically, how does this variability impact the development of antiviral treatments?
Line 77-81: The interaction between viral E proteins and cell receptors is mentioned but not elaborated upon. Given that this is a review focused on antiviral molecules, a discussion on how these interactions could be targeted for antiviral treatments would be beneficial.
Minor Comments:
Line 68-72: The figure description is informative but could be enhanced by specifying how this information is relevant to the paper's focus on antiviral molecules.
Line 84-92: The sentence structure and flow could be improved for better readability. Consider breaking down the long sentences into shorter, more focused statements.
Line 80-81: The table is introduced abruptly. A brief introduction to what the table aims to show would improve the flow of the text.
3. Transmission and Pathology
Major Comments:
Line 95-100: The section provides a general overview of flavivirus transmission and symptoms. However, it does not explicitly link this information to the paper's main focus on antiviral molecules. Consider discussing how understanding transmission and pathology could inform antiviral strategies.
Line 102-113: The paper discusses various factors affecting the severity of flavivirus infections. This is an important point that could be further elaborated upon. Specifically, how do these factors complicate the development of universal antiviral treatments?
Minor Comments:
Line 97-100: The mention of severe hemorrhagic fever or neurological damage as potential outcomes is significant but could be strengthened by providing statistics or references to case studies.
Line 106-113: The discussion on alternative transmission routes is informative but could be enhanced by discussing its implications for antiviral treatments.
4. Viral Targets and Drugs with Antiviral Activity
Major Comments:
Line 116-118: The text appropriately emphasizes the need for stable targets in antiviral treatments. However, it would be beneficial to elaborate on why this stability is particularly crucial for flaviviruses, given their propensity for mutation and recombination.
Line 119-121: The mention of Hepatitis C Virus (HCV) and its quasi-species formation is pertinent but somewhat tangential to the main focus on flaviviruses. Consider rephrasing to directly relate this phenomenon to flaviviruses and their treatment challenges.
Line 122-127: While the section promises an analysis of viral targets in different stages of the replication cycle, it does not delve into specifics. The reader would benefit from a detailed discussion or a table summarizing these targets, their associated antiviral molecules, and the current state of research or clinical trials concerning them.
Minor Comments:
Line 124-125: The comparison with HIV and HCV drug development is useful but would be more informative if it included specific examples of how these strategies could be adapted for flaviviruses.
Line 127: The sentence ends with a promise to analyze viral targets in different stages of the replication cycle. This could be a segue into a new subsection that provides the detailed analysis, thereby enhancing the structural organization of the paper.
Line 118-119: The term "Viral RdRp" is introduced without explanation. Given the advanced reader comprehension level assumed, it may not require a full definition, but a brief clarification could improve readability.
In summary, the section on "Viral Targets and Drugs with Antiviral Activity" sets the stage for a detailed discussion on antiviral strategies against flaviviruses. However, it currently lacks the depth and specificity that would make it a valuable contribution to the field. Expanding on the points mentioned, particularly with regard to the unique challenges posed by flaviviruses, would significantly enhance the section's scholarly impact.
5. Entry Inhibitors and Subsequent Sections
Major Comments:
Line 131-146: The section provides a comprehensive overview of the mechanisms by which flaviviruses enter host cells. However, it would be beneficial to include a diagram or flowchart to visually represent these complex interactions and stages.
Line 153-160: The comparison with HIV treatment is insightful but could be misleading. Clarifying why these molecules are not as effective against flaviviruses as they are against HIV would provide a more nuanced understanding.
Line 169-174: The introduction to synthetic peptides is well-written but could benefit from a table summarizing the peptides discussed, their targets, and their efficacy in both in vitro and in vivo models.
Line 209-233: The section on monoclonal antibodies is informative but could be enhanced by discussing the challenges and limitations of using mAbs for flavivirus treatment, especially in the context of antibody-dependent enhancement (ADE).
Line 235-294: The discussion on synthetic and natural compounds and host function targets is extensive but could be more organized. Consider breaking it down into sub-sections based on the type of compounds and their mechanisms of action.
Minor Comments:
Line 137-138: The term "clathrin-rich regions" is introduced without explanation. A brief description or footnote would be helpful for readers unfamiliar with the term.
Line 176-180: The Z2 peptide's efficacy is mentioned, but it would be helpful to know if there are any associated side effects or limitations.
Line 265-275: The anticancer drugs erlotinib and sunitinib are mentioned, but their limitations in crossing the blood-brain barrier should be highlighted more prominently as it's a significant drawback for flavivirus treatment.
Line 287-288: The failure of chloroquine in phase 2 trials is crucial and should be discussed in more detail, including theories or evidence as to why it failed.
Line 292-294: The multi-target action of Niclosamide is intriguing and warrants further discussion or speculation on its potential as a broad-spectrum antiviral agent.
In summary, the section on "Entry Inhibitors" and subsequent sections are comprehensive and informative but could benefit from better organization and the inclusion of visual aids. Addressing these points would make the paper a more valuable resource for researchers in the field.
section on viral replication, polyprotein synthesis, and processing in flaviviruses. Here are my comments:
Overall Assessment:
The section provides a comprehensive overview of the molecular mechanisms involved in flavivirus replication, focusing on the role of various viral and host proteins. It also delves into potential antiviral targets and existing drugs that could inhibit these processes. The text is generally well-written and informative but could benefit from some improvements for clarity and completeness.
Specific Comments:
Clarity and Structure: The section is dense with information, which is good for a scientific audience but could be overwhelming for a general reader. Consider breaking down complex sentences and using subheadings for different aspects like "Role of NS3," "Potential Drug Targets," etc.
Terminology: The text uses a lot of specialized terms and acronyms without always providing sufficient context or definitions. It would be helpful to define these terms when they are first introduced.
Citations: The text is well-cited, which adds credibility. However, it would be beneficial to include a brief summary of the key findings from the cited works to provide context.
Mechanistic Details: While the section does a good job of listing the proteins involved in flavivirus replication and potential drug targets, it could benefit from more mechanistic details. For example, how exactly does NS4A contribute to replication organelle formation?
Drug Targets and Efficacy: The section covers a wide range of potential antiviral drugs and their targets. However, it would be beneficial to summarize the current state of these drugs in clinical trials or their stage of development.
Grammar and Syntax: There are some minor grammatical errors and awkward phrasings that could be smoothed out for better readability.
Figures and Tables: Consider including figures or tables to summarize the complex interactions between different proteins and potential drug targets.
Future Directions: The section could benefit from a brief discussion on the future directions of this research field, including any upcoming drugs or therapies that are in the pipeline.
Conclusions: A summary paragraph at the end of the section could help in consolidating the key points and giving the reader a take-home message.
Recommendations for Improvement:
Add definitions for specialized terms and acronyms.
Include brief summaries of key findings from cited works.
Standardize units of concentration for drug efficacy.
Add more mechanistic details where possible.
Include a summary paragraph and a section on future directions.
Overall, the section is informative and covers a lot of ground but could be improved in terms of clarity, detail, and structure.
Section 7: Assembly and Egress Inhibitors
Clarity and Organization: The section is well-organized and provides a comprehensive overview of the assembly and egress inhibitors targeting flaviviruses. However, the transition between the cellular mechanisms and the specific inhibitors could be smoother.
Scientific Rigor: The section is well-cited and appears to be based on rigorous scientific research. It would be beneficial to include more recent studies if available.
Specific Comments:
Line 608-609: Consider elaborating on the "acidified compartment" where furin cleaves prM.
Line 618-620: The term "Defective processing of N-linked oligosaccharides" could be explained in simpler terms for broader understanding.
Line 627: The dismissal of the celgosivir trial is mentioned but not why it was dismissed. Was it due to safety concerns or ineffectiveness?
Recommendations:
Consider adding a summary sentence at the end to encapsulate the potential and limitations of targeting assembly and egress for antiviral strategies.
Section 7.1: C Protein Targeting
Clarity and Organization: The section is clear but could benefit from a brief introduction to set the context for C protein targeting.
Scientific Rigor: The section is well-cited, but the mechanisms of action for the molecules like VGTI-A3 could be described in more detail.
Specific Comments:
Line 666-668: The description of VGTI-A3 is a bit technical. Consider simplifying the language.
Line 674: The term “kissing” binding could be confusing. Consider providing a brief explanation or alternative terminology.
Recommendations:
A summary sentence highlighting the promise and challenges of C protein targeting could enhance the section.
Section: Trials
Clarity and Organization: The section is mostly clear but could benefit from introductory sentences for each type of trial or drug class.
Scientific Rigor: The section is well-cited and provides a good overview of ongoing and completed trials. However, the status of some trials is not clear.
Specific Comments:
Line 717-719: The use of the term "molecules" could be replaced with "compounds" or "agents" for clarity.
Line 760: The reasons for discontinuation of some studies could be elaborated upon.
Recommendations:
Consider adding a concluding paragraph summarizing the current landscape of clinical trials and what they imply for future antiviral strategies against flaviviruses.
General Recommendations
A glossary of technical terms and abbreviations could make the manuscript more accessible to a broader audience.
Consider including figures or tables to summarize the mechanisms of action of different antiviral strategies.
Overall, the manuscript sections provide a thorough and well-cited overview of the current state of antiviral strategies against flaviviruses. With some minor revisions for clarity and completeness, it could serve as a valuable resource in the field.
Section: Conclusions
Clarity and Organization: The conclusion is well-structured and provides a comprehensive summary of the key points discussed in the manuscript. It effectively ties together the various aspects of flavivirus antiviral strategies.
Scientific Rigor: The section does a good job of summarizing the current state of research in the field. However, it could benefit from citations to support some of the statements made, especially concerning the potential for multi-drug combinations.
Specific Comments:
Line 765-767: The opening sentence is a bit long and could be broken down for better readability.
Line 772-774: The mention of "synthetic peptides and putative antiviral molecules revealed through in silico studies" could be expanded upon or cited.
Line 779-780: The statement about studies not confirming in vitro results or being discontinued could be more specific. Are these studies not confirming results due to methodological issues, or is it more about the complexity of in vivo systems?
Line 785-789: The discussion about multi-drug combinations is intriguing but could be strengthened with references to studies that have successfully employed this approach in other viral infections.
Recommendations:
Consider adding a few sentences that discuss the future directions of research in flavivirus antiviral strategies. What are the next steps, and what should researchers focus on?
A brief mention of the limitations of current research could provide a more balanced view.
It may be useful to include a call to action for the scientific community, emphasizing the importance of continued research in this area.
General Recommendations
A final summary sentence could help to encapsulate the main takeaways from the conclusion and the manuscript as a whole.
Consider including a brief section on the implications of this research for public health, especially in regions most affected by flavivirus infections.
Overall, the conclusion does an excellent job of summarizing the complex landscape of antiviral strategies against flaviviruses. With minor adjustments for clarity and the inclusion of additional citations, this section could serve as a strong closing statement for the manuscript.
Overall, the English language usage in the manuscript is generally clear and understandable. However, there are some areas where improvements could be made to enhance readability and clarity:
1. Sentence Structure: Some sentences are quite long and could be broken down into shorter, more digestible segments. This would make the text easier to read and understand. For example, the opening sentence in the conclusion section could be simplified.
2. Grammar and Syntax: The manuscript is mostly free of grammatical errors, but a thorough proofreading could help catch minor issues that may have been overlooked.
3. Consistency: Ensure that tense and voice are consistent throughout the manuscript. Switching between past and present tense can be confusing for the reader.
4. Terminology: The manuscript uses specialized terminology appropriately, but it would be beneficial to ensure that all terms are defined when first introduced, especially for the benefit of readers who may not be experts in the field.
5. Clarity: While the manuscript is generally well-written, some statements could be clarified for better understanding. For instance, the discussion about multi-drug combinations in the conclusion could be more explicit.
6. Punctuation: Pay attention to the use of commas and semicolons to break up long sentences and lists. Proper punctuation can significantly improve the flow of the text.
I recommend a thorough proofreading and possibly consultation with a native English speaker familiar with the scientific context to polish the manuscript further. Overall, the language quality is good but could benefit from minor revisions for clarity and readability.
Author Response

(The authors gave the same response as above.)
